# MODEL EXTRAPOLATION EXPEDITES ALIGNMENT

## ABSTRACT

As the alignment training of large language models (LLMs) usually requires expensive computational resources, exploring more efficient alignment methods to reduce training overhead has always been an important and compelling research challenge. Inspired by prior work on *model interpolation*, we present a simple method called **ExPO** (*model extrapolation*) to expedite the alignment of LLMs with human preferences. Based on our observation that interpolating the weights between existing DPO/RLHF models and their initial SFT checkpoints usually produces new models with intermediate performance, we propose to treat a partially-trained model $\mathcal{M}_1$ (corresponding to the intermediate-performing model) as the interpolated result between the initial SFT checkpoint $\mathcal{M}_0$ and a hypothetical better-aligned model $\mathcal{M}_2$. Thus, we can obtain the hypothetical $\mathcal{M}_2$ by simply extrapolating the model weights along the direction from $\mathcal{M}_0$ to $\mathcal{M}_1$, which consequently saves the additional training overhead for $\mathcal{M}_1$ to reach better alignment performance. We validate our hypothesis through controlled experiments, demonstrating that ExPO can boost a DPO model trained with only 20% steps to outperform the fully-trained one. Additionally, we show that ExPO can also notably improve existing open-source LLMs (ranging from 1.8B to 70B parameters), as evidenced by evaluations on the mainstream LLM benchmarks AlpacalEval 2.0 and MT-Bench, which further highlights ExPO's utility and potential in enabling more efficient LLM alignment.

## 1 INTRODUCTION

Large language models (LLMs) typically require additional fine-tuning to learn to follow human instructions after unsupervised pre-training on massive textual corpora (OpenAI, 2022; 2023; Bai et al., 2022). The current fine-tuning paradigm consists of two steps: supervised fine-tuning (SFT) and human preference optimization. SFT employs a similar language modeling objective to pre-training, where the model is trained to maximize the likelihood of responses on high-quality demonstration data. Human preference optimization, on the other hand, aims to adjust the model's response distribution to better *align with human preferences*. However, the alignment training[1] process, as exemplified by the well-known Reinforcement Learning from Human Feedback (RLHF; Ouyang et al. 2022; Schulman et al. 2017) and Direct Preference Optimization (DPO; Rafailov et al. 2023), still requires expensive computational resources (Ji et al., 2024a; Meng et al., 2024). This underscores the significance of exploring more efficient alignment methods to reduce the training overhead.

Our work draws inspiration from the literature on *model interpolation* (or model averaging). This technique leverages mode connectivity of neural networks (Garipov et al., 2018; Entezari et al., 2022) and interpolates model weights between multiple fine-tuned models (e.g., trained with different initializations or data subsets) to improve out-of-distribution generalization (Izmailov et al., 2018; Lin et al., 2024; Wortsman et al., 2022). It has been commonly adopted in recent LLMs like Gemma-2 (Gemma et al., 2024) and LLaMA-3 (Dubey et al., 2024). Given our interest in the alignment training, we applied model interpolation to existing open-source DPO/RLHF models (Tunstall et al., 2023; Cai et al., 2024; Zhu et al., 2023) and their initial SFT checkpoints. Interestingly, we observed that model interpolation, while producing new models that can generate normal responses, usually results in in-between performance compared to the original models, as shown in Figure 1.

---

[1]In this paper, we use the term "alignment training" to refer to the process of *training LLMs to align with human preferences*, such as via the RLHF or DPO algorithms.

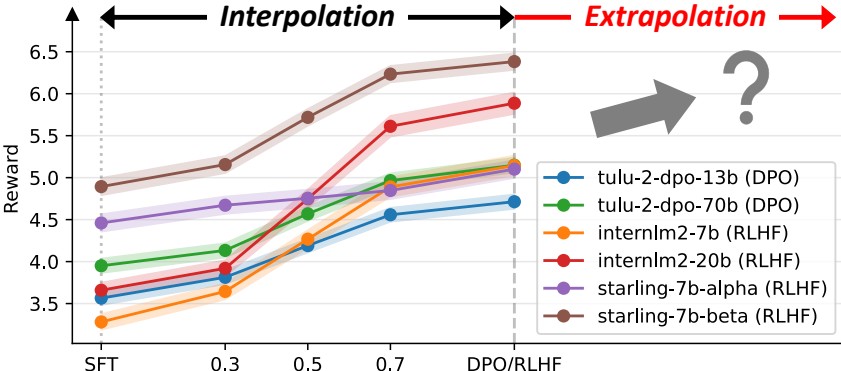

Figure 1: Model interpolation usually produces a new model with intermediate performance between the DPO/RLHF model and the initial SFT checkpoint. This observation motivates our proposal of the interpolation hypothesis and the derived ExPO method. Reward scores (§ 3.1) are calculated on the UltraFeedback (Cui et al., 2023) development set.

Intrigued by the observation from model interpolation, we propose an interesting but unexplored hypothesis: *A partially-trained model $\mathcal{M}_1$ (e.g., with fewer training steps) may be treated as the interpolated result between its initial SFT checkpoint $\mathcal{M}_0$ and a better-aligned model $\mathcal{M}_2$ that requires more training steps to achieve.* In other words, we assume that in the parameter space, $\mathcal{M}_1$ lies on the linear path from $M_0$ to a hypothetically existing, better-aligned $M_2$. Thereby, we can simply obtain the hypothetical $\mathcal{M}_2$ by reversely *extrapolating* the model weights along the direction from $\mathcal{M}_0$ to $\mathcal{M}_1$, as indicated by the gray arrow in Figure 1. If this hypothesis holds, we can bypass further training of $\mathcal{M}_1$, and instead directly reach better alignment performance (corresponding to $\mathcal{M}_2$) with remarkably reduced training overhead.

We refer to the process of obtaining $\mathcal{M}_2$ as **ExPO (*model extrapolation*)**. To empirically validate the aforementioned hypothesis and ExPO's effectiveness, we conduct controlled experiments using HuggingFace's official checkpoints and training recipe. We first demonstrate that ExPO notably boosts the DPO models using fewer training steps (e.g., only 20%) to outperform the fully-trained one (§ 3.2), with the improvement of up to 8.4% on AlpacalEval 2.0 (Li et al., 2023). We then conduct ablation studies to identify key factors influencing ExPO's efficacy, including training data quality (§ 3.4) and training configurations such as training hyperparameters and optimizer (§ 3.5). Furthermore, we extend ExPO's application to twelve open-source LLMs (§ 4.1), ranging from 1.8B to 70B parameters, which have undergone varied alignment training such as offline DPO, iterative DPO, or online RLHF. We show that ExPO consistently improves these LLMs, by up to 4.5% on AlpacaEval 2.0 and 0.37 on MT-Bench (Zheng et al., 2023b), which suggests that ExPO can also serve as a practical and efficient means to compensate for potential inadequate training of existing, already-aligned LLMs. In summary, our work demonstrates the utility of model extrapolation in enabling more efficient LLM alignment, which may inspire follow-up studies and broader applications in future research.

## 2 METHODOLOGY

Our proposed ExPO method is inspired by the observation from model interpolation and builds upon the aforementioned interpolation hypothesis. Formally, we denote the language model's parameter space as $\Theta$ and suppose that the alignment performance can be quantified by a continuous scalar function $\Omega : \Theta \to \mathbb{R}$, where higher $\Omega(\boldsymbol{\theta})$ indicates better alignment with human preferences. We suppose that the model $\mathcal{M}_1$ (parameterized by $\boldsymbol{\theta}_1$) has undergone moderate alignment training (e.g., via DPO). We denote its SFT checkpoint as $\mathcal{M}_0$ (parameterized by $\boldsymbol{\theta}_0$), which is used for initializing $\mathcal{M}_1$ and satisfies $\Omega(\boldsymbol{\theta}_0) < \Omega(\boldsymbol{\theta}_1)$.

**Interpolation Hypothesis** We hypothesize that there exists a better-aligned model $\mathcal{M}_2$ (parameterized by $\boldsymbol{\theta}_2$) that satisfies $\Omega(\boldsymbol{\theta}_1) < \Omega(\boldsymbol{\theta}_2)$, and an interpolation coefficient $\gamma \in [0, 1]$ such that:

$$\boldsymbol{\theta}_1 = (1 - \gamma)\boldsymbol{\theta}_0 + \gamma\boldsymbol{\theta}_2. \tag{1}$$

In other words, $\mathcal{M}_1$ is assumed to lie on the linear path from $\mathcal{M}_0$ toward some better-aligned $\mathcal{M}_2$. Note that the current hypothesis takes the simplest form of uniform interpolation, i.e., using the same interpolation coefficient for all model modules, which can be extended to more sophisticated forms in future work.

**ExPO: Model Extrapolation** Deriving from the above hypothesis, we can simply obtain the hypothetical $\mathcal{M}_2$ by *extrapolating* the weights along the direction from $\mathcal{M}_0$ to $\mathcal{M}_1$, as illustrated in Figure 2. With the substitution of $\alpha = 1/\gamma - 1 \in [0, +\infty)$, the process of model extrapolation, or ExPO, can be formulated as follows:

$$\boldsymbol{\theta}_2 = (1 + \alpha)\boldsymbol{\theta}_1 - \alpha\boldsymbol{\theta}_0 = \boldsymbol{\theta}_1 + \alpha(\boldsymbol{\theta}_1 - \boldsymbol{\theta}_0) = \boldsymbol{\theta}_1 + \alpha\triangle\boldsymbol{\theta}, \quad (2)$$

which can be easily implemented within just a few lines of code. In practice, the coefficient $\alpha$ acts as the hyperparameter that controls the extrapolation length. Note that the search for $\alpha$ only involves model inference. Given the rapid development of high-performance LLM inference infrastructures, such as vLLM (Kwon et al., 2023) and SGLang (Zheng et al., 2023c), the search process consumes remarkably less GPU hardware resources and GPU time compared to model training, which can ideally save additional training overhead for $\mathcal{M}_1$ to reach better alignment performance.

Figure 2: The **solid orange curve** indicates the training trajectory from $\boldsymbol{\theta}_0$ to $\boldsymbol{\theta}_1$, while the **dashed orange line** denotes the extrapolation along the direction of $\triangle\theta$, thus producing $\boldsymbol{\theta}_2$.

In the following § 3, we will conduct controlled experiments to empirically validate the interpolation hypothesis and identify key factors influencing ExPO's efficacy, including training data quality and detailed training configurations.

## 3 CONTROLLED EXPERIMENTS

### 3.1 SETUP AND EVALUATION PROTOCOL

**Model and Training Recipe** To conduct controlled experiments, we follow the training recipe of the `zephyr-7b-dpo` model that is officially released by HuggingFace[2]. This open training recipe, along with the resulting checkpoints, allows us to adjust the training data and configurations as desired, and they are also widely used in controlled experiments in recent LLM alignment research (Chen et al., 2024b; Ji et al., 2024b; Chen et al., 2024a).

Specifically, we use the same UltraFeedback (Cui et al., 2023) dataset for model training, which is a classical preference dataset that is popularly used for LLM alignment (Ivison et al., 2023; Tunstall et al., 2023; Zhu et al., 2023; Dong et al., 2024). UltraFeedback contains diverse instruction-response pairs with GPT-4-annotated preference labels, split into 61K and 1K data as the training and development sets, respectively. For DPO training, we use `zephyr-7b-dpo`'s SFT checkpoint, `zephyr-7b-sft`, for model initialization and as the reference model. We adopt the same global batch size of 128, the learning rate of 5e-7, the AdamW (Loshchilov & Hutter, 2019) optimizer, and the DPO hyperparameter $\beta$ of 0.01. While `zephyr-7b-dpo` is trained for one epoch (478 optimization steps in total), in § 3.2 we will vary the training steps to investigate ExPO's effectiveness.

To determine the optimal $\alpha$ value in ExPO, we use a combination of binary search and grid search with manually tuned intervals. We select the $\alpha$ giving the highest expected reward on the UltraFeedback development set (1K instructions), as calculated by an open-source reward model[3].

**Evaluation Protocol** We resort to **AlpacaEval 2.0** (Li et al., 2023) for model evaluation, which is a leading benchmark that assesses LLMs' instruction-following ability and their alignment with human preferences. It contains a fixed set of 805 instructions chosen to be representative of real user cases. For each instruction, it calculates the probability that a GPT-4 Turbo evaluator prefers the output of the evaluated model over the GPT-4 baseline, thus providing an affordable and replicable alternative to human annotation. The **win rate** over the GPT-4 baseline is computed as the expected preference probability. Recently, AlpacaEval 2.0 has introduced the new **length-controlled (LC)**

---

[2]`https://github.com/huggingface/alignment-handbook/blob/main/recipes/`
`zephyr-7b-beta/dpo/config_full.yaml`
[3]`https://huggingface.co/weqweasdas/RM-Mistral-7B`

**win rate** metric (Dubois et al., 2024), which alleviates the length bias of the GPT-4 Turbo evaluator (i.e., the prior preference toward longer responses) and has a Spearman correlation of 0.98 with the real-world human evaluation on Chatbot Arena (Zheng et al., 2023b).

In § 3.2, we report both the basic and LC win rates, as well as the expected reward over the 805 instructions calculated by the reward model. For the ablation studies in § 3.4 and 3.5, we calculate the expected reward on the UltraFeedback development set (1K instructions) for ease of analysis.

## 3.2 RESULTS OF VARYING TRAINING STEPS

Initialized from the SFT checkpoint ($\mathcal{M}_0$), we first train the DPO models ($\mathcal{M}_1^*$) with varying steps (10%, 20%, 40%, and 100%), where all other training configurations are kept the same. We then apply ExPO to obtain the extrapolated model $\mathcal{M}_2^*$. The evaluation results in Table 1 show that, although fewer training steps typically result in lower-tier performance, ExPO can effectively bridge the gap caused by the reduced training steps. For instance, in terms of the LC win rate, ExPO boosts $\mathcal{M}_1^{10\%}$ by 5.8% (to $\mathcal{M}_2^{10\%}$'s **16.3%**) and $\mathcal{M}_1^{20\%}$ by 8.4% (to $\mathcal{M}_2^{20\%}$'s **21.3%**), making them perform comparably and surpass the fully-trained $\mathcal{M}_1^{100\%}$ (17.3%), respectively.

Table 1: Evaluation results on AlpacaEval 2.0 of applying ExPO to DPO models trained with varying steps ($\mathcal{M}_1^*$). All the DPO models ($\mathcal{M}_1^*$) are initialized from the SFT checkpoint `zephyr-7b-sft` ($\mathcal{M}_0$). We directly use `zephyr-7b-dpo` as the DPO model trained with 100% steps ($\mathcal{M}_1^{100\%}$).

| | Reward | Win Rate | LC Win Rate |
|---|---|---|---|
| SFT ($\mathcal{M}_0$) | 3.42 | 4.7% | 8.7% |
| DPO, 10% training steps ($\mathcal{M}_1^{10\%}$) | 3.97 | 5.9% | 10.4% |
| + ExPO ($\mathcal{M}_2^{10\%}$) | 6.57 (+2.60) | 17.9% (+12.0%) | 16.3% (+5.8%) |
| DPO, 20% training steps ($\mathcal{M}_1^{20\%}$) | 4.70 | 8.6% | 12.9% |
| + ExPO ($\mathcal{M}_2^{20\%}$) | **6.95** (+2.25) | **22.7%** (+14.2%) | **21.3%** (+8.4%) |
| DPO, 40% training steps ($\mathcal{M}_1^{40\%}$) | 5.77 | 12.1% | 14.6% |
| + ExPO ($\mathcal{M}_2^{40\%}$) | 6.75 (+0.98) | 17.7% (+5.6%) | 16.6% (+2.0%) |
| DPO, 100% training steps ($\mathcal{M}_1^{100\%}$) | 6.16 | 14.7% | 17.3% |
| + ExPO ($\mathcal{M}_2^{100\%}$) | 6.52 (+0.36) | 18.0% (+3.3%) | 20.2% (+2.8%) |

In terms of *computational overhead*, training $\mathcal{M}_1^{100\%}$ takes about 12 GPU hours (on A100 80GB). In contrast, for $\mathcal{M}_2^{20\%}$, the hyperparameter search process of ExPO takes less than 0.5 GPU hours. Adding $\mathcal{M}_1^{20\%}$'s training, which takes about 2.5 GPU hours, ExPO results in a **75% reduction** in computational overhead while achieving same-tier (or superior) alignment performance. Since ExPO's hyperparameter search only involves model inference, it also remarkably reduces the GPU hardware requirements compared to model training (e.g., for a 7B model, a single A10 24GB vs. eight A100 80GB). These empirical results validate the soundness of our interpolation hypothesis in § 2 and demonstrate that ExPO can effectively expedite LLM alignment.

**Other Observations** Meanwhile, we observe several other noteworthy phenomena:

- The reward score is not perfectly correlated with the LC win rate, as seen in the comparison between $\mathcal{M}_2^{40\%}$ and $\mathcal{M}_2^{100\%}$. We believe this is a limitation of our currently used reward model. Employing stronger reward models in the future could lead to more accurate evaluation results and better hyperparameter search outcomes.

- For $\mathcal{M}_1^{100\%}$, applying ExPO ($\mathcal{M}_2^{100\%}$) still brings moderate performance improvement (by 2.8%). This suggests that ExPO may enhance more existing, already-aligned LLMs, and we will extend ExPO's application in § 4.1.

- The model performance after applying ExPO does not simply improve with increased training steps (e.g., $\mathcal{M}_2^{20\%}$ outperforms $\mathcal{M}_2^{100\%}$). This implies that the effectiveness of ExPO may be influenced by other key factors, e.g., training data and detailed training configurations. We will conduct ablation studies in § 3.4 and 3.5 to analyze these factors.

### 3.3 WHY EXPO CAN WORK?

Given the surprising effectiveness of EXPO shown in § 3.2, we attempt to discuss why EXPO can work before proceeding with more empirical studies. Under the formulation of § 2, EXPO can be explained as *a first-order approximation that implicitly optimizes the alignment objective* $\Omega$. By applying a first-order Taylor Expansion of $\Omega$ at $\boldsymbol{\theta}_1$, we get:

$$\Omega(\boldsymbol{\theta}_2) = \Omega(\boldsymbol{\theta}_1 + \alpha\Delta\boldsymbol{\theta}) \approx \Omega(\boldsymbol{\theta}_1) + \alpha\nabla\Omega(\boldsymbol{\theta}_1) \cdot \Delta\boldsymbol{\theta}. \tag{3}$$

For this approximation to hold, the condition $|\alpha\Delta\boldsymbol{\theta}|$ needs to be small. Interestingly, we observe that this condition aligns with the hyperparameter search results in § 3.2 (Appendix § B). Specifically, for $\mathcal{M}_2^{10\%}$, $\mathcal{M}_2^{20\%}$, $\mathcal{M}_2^{40\%}$, and $\mathcal{M}_2^{100\%}$, the optimal $\alpha$ values found are 8.0, 2.5, 0.5, and 0.3, respectively. This suggests that models trained with more steps, whose $|\Delta\boldsymbol{\theta}|$ is usually larger, require smaller optimal $\alpha$ values. Based on this consistency, we can reasonably assume that the first-order approximation holds in the following discussion.

EXPO's effectiveness indicates that $\Omega(\boldsymbol{\theta}_2) > \Omega(\boldsymbol{\theta}_1)$, or equivalently, $\nabla\Omega(\boldsymbol{\theta}_1) \cdot \Delta\boldsymbol{\theta} > 0$. In other words, the gradient of $\Omega$ at $\boldsymbol{\theta}_1$ has a positive component along the direction of $\Delta\boldsymbol{\theta}$, which can be intuitively illustrated by Figure 2. We thus speculate that for EXPO to work in practice, two requirements need to be satisfied: **First**, $\boldsymbol{\theta}_1$ should not be the local optimum or an overfitted point of $\Omega$. **Second**, $\Delta\boldsymbol{\theta}$ should indicate a direction that genuinely improves alignment performance[4], rather than exploiting spurious features (e.g., length bias; Park et al. 2024). We will discuss the two requirements more specifically in conjunction with the subsequent ablation studies in § 3.4 and 3.5.

### 3.4 ANALYSIS OF TRAINING DATA QUALITY

In § 3.2, we observed that the model performance after applying EXPO does not simply improve with the increased training steps. We speculate that this may be because increasing the training steps makes the model more prone to learning spurious features from the training data, such as length bias[5]. This issue consequently prevents $\Delta\boldsymbol{\theta}$ from indicating a direction that genuinely improves alignment performance, corresponding to the second requirement in § 3.3.

Table 2: Ablation results on UltraFeedback (development set) of adjusting the training data quality. "N/A" denotes that the calculated expected reward does not improve after applying EXPO even with the smallest $\alpha = 0.1$.

| Training Data | Original ($\mathcal{M}_1^*$) | | + EXPO ($\mathcal{M}_2^*$) | | |
|---|---|---|---|---|---|
| | Reward | Length | Optimal $\alpha$ | Reward | Length |
| 10% training steps, random ($\mathcal{M}_*^{10\%}$) | 3.59 | 262 | 8.0 | **5.82** | 541 |
| 10% training steps, length-biased ($\mathcal{M}_*^{10\%,b}$) | 4.62 | 770 | 0.2 | 4.69 | 810 |
| 20% training steps, random ($\mathcal{M}_*^{20\%}$) | 4.37 | 294 | 2.5 | **6.08** | 567 |
| 20% training steps, length-biased ($\mathcal{M}_*^{20\%,b}$) | 5.05 | 748 | 0.4 | 5.11 | 875 |
| 40% training steps, random ($\mathcal{M}_*^{40\%}$) | 5.30 | 407 | 0.5 | **5.80** | 594 |
| 40% training steps, length-biased ($\mathcal{M}_*^{40\%,b}$) | 4.90 | 671 | N/A | N/A | N/A |

To analyze the impact of training data quality on EXPO's efficacy in a controlled manner, we take *length bias* as an example and investigate changes in model performance by artificially introducing length bias into the training data. Specifically, unlike the random sampling of training data in § 3.2, here we arrange the training data in descending order based on the length difference between preferred and unpreferred responses. The evaluation results in Table 2 show that, although introducing length bias can temporarily improve the reward score ($\mathcal{M}_1^{10\%,b}$ and $\mathcal{M}_1^{20\%,b}$ outperform $\mathcal{M}_1^{10\%}$ and $\mathcal{M}_1^{20\%}$, respectively), the performance after applying EXPO is consistently worse ($\mathcal{M}_2^{10\%,b}$

---

[4]It is particularly worth noting that Equation 2 can be rewritten as $\boldsymbol{\theta}_2 = \boldsymbol{\theta}_0 + (1 + \alpha)\Delta\boldsymbol{\theta}$. Therefore, the controlled setup in § 3 makes it possible for us to analyze the role of $\Delta\boldsymbol{\theta}$, as with the fixed $\mathcal{M}_0$, the achievable performance of $\mathcal{M}_2$ is only dependent on $\Delta\boldsymbol{\theta}$.

[5]The average lengths of preferred and unpreferred responses in the UltraFeedback training set are 319 and 277 tokens, respectively.

and $\mathcal{M}_2^{20\%,\mathrm{b}}$ underperform $\mathcal{M}_2^{10\%}$ and $\mathcal{M}_2^{20\%}$, respectively). Moreover, we find that the optimal $\alpha$ values corresponding to $\mathcal{M}_2^{10\%,\mathrm{b}}$ and $\mathcal{M}_2^{20\%,\mathrm{b}}$ are 0.2 and 0.4, respectively, which are significantly smaller than those for $\mathcal{M}_2^{10\%}$ and $\mathcal{M}_2^{20\%}$ (8.0 and 2.5, respectively), and $\mathcal{M}_1^{40\%,\mathrm{b}}$ even fails to show improvement after applying ExPO. These results suggest that the lowered training data quality, e.g., with length bias, can cause $\Delta\boldsymbol{\theta}$ to fail in indicating a genuine direction of alignment optimization, thus impairing the achievable performance by ExPO.

## 3.5 ANALYSIS OF TRAINING CONFIGURATIONS

We next analyze the impact of detailed training configurations on ExPO's efficacy. Specifically, since ExPO amplifies the weight changes $\Delta\boldsymbol{\theta}$ from $\mathcal{M}_0$ to $\mathcal{M}_1$, we are interested in whether ExPO is equivalent to directly increasing the magnitude of weight changes, including through increasing the **training epochs** or **learning rate**. Additionally, we investigate how configurations closely related to the training trajectory from $\mathcal{M}_0$ to $\mathcal{M}_1$, such as the hyperparameter $\beta$ **in DPO** and the choice of **optimizer**, influence ExPO's effectiveness.

**Training Epochs and Learning Rate**  Using the same training data and configurations as in the setup with 20% training steps in § 3.2, we investigate whether ExPO is equivalent to directly increasing the training epochs or learning rate to increase the magnitude of weight changes. The evaluation results in Table 3 show that, both $\mathcal{M}_1$ and $\mathcal{M}_2$ after increasing the training epochs or learning rate perform worse compared to the default configuration of $\mathcal{M}_2$, while the optimal $\alpha$ values for the former $\mathcal{M}_2$ are also much smaller than for the latter. This suggests that increasing the training epochs or learning rate may more easily lead to overfitting or cause $\Delta\boldsymbol{\theta}$ to fail in indicating a genuine direction of alignment optimization, corresponding to the two requirements in § 3.3.

Table 3: Ablation results on UltraFeedback of increasing the training epochs or learning rate.

| Training Epochs | $\mathcal{M}_1$ | $\mathcal{M}_2$ | | Learning Rate | $\mathcal{M}_1$ | $\mathcal{M}_2$ | |
|---|---|---|---|---|---|---|---|
| | Reward | $\alpha$ | Reward | | Reward | $\alpha$ | Reward |
| 1 ($\times$1; default) | 4.37 | 2.5 | **6.08** | 5e-7 ($\times$1; default) | 4.37 | 2.5 | **6.08** |
| 2 ($\times$2) | 4.93 | 0.3 | 5.06 | 1e-6 ($\times$2) | 5.20 | 0.5 | 5.54 |
| 3 ($\times$3) | 4.47 | N/A | N/A | 2e-6 ($\times$3) | 5.33 | 0.4 | 5.52 |

$\beta$ **in DPO and Optimizer**  We also adjust the hyperparameter $\beta$ in DPO, which controls the strength of the KL constraint in the DPO algorithm, as well as the optimizer used for training $\mathcal{M}_1$. We adjust $\beta$ among 0.001, 0.01 (default), and 0.1, and the optimizer among AdamW (defualt), AdaGrad (Duchi et al., 2011), and RMSprop (Hinton, 2012), to ensure a diverse range of ablation studies. Note that we study the two configurations because they intuitively influence the training trajectory from $\mathcal{M}_0$ to $\mathcal{M}_1$ and the resulting $\Delta\boldsymbol{\theta}$. From the left part of Table 4, we observe that applying ExPO to $\mathcal{M}_1$ trained with different $\beta$ values leads to consistent improvements, and the performance across different $\beta$ values is similar (for both $\mathcal{M}_1$ and $\mathcal{M}_2$). This may be because, given the same other configurations (e.g., the seen training data, learning rate, etc.), DPO training with different $\beta$ values tends to stably converge to a similar region.

Table 4: Ablation results on UltraFeedback of varying $\beta$ in DPO and training optimizers.

| $\beta$ **in DPO** | $\mathcal{M}_1$ | $\mathcal{M}_2$ | | Optimizer | $\mathcal{M}_1$ | $\mathcal{M}_2$ | |
|---|---|---|---|---|---|---|---|
| | Reward | $\alpha$ | Reward | | Reward | $\alpha$ | Reward |
| 0.01 (default) | 4.37 | 2.5 | 6.08 | AdamW (default) | 4.37 | 2.5 | 6.08 |
| 0.1 | 4.36 | 2.5 | 6.43 | AdaGrad | 3.42 | 15.0 | 6.25 |
| 0.001 | 4.31 | 3.0 | 6.34 | RMSprop | 4.88 | 0.4 | 5.08 |

On the other hand, from the right part of Table 4, we observe a more interesting phenomenon. Specifically, although AdaGrad converges more slowly, i.e., the $\mathcal{M}_1$ performs worst, its $\mathcal{M}_2$ achieves slightly higher performance than AdamW. In contrast, while RMSprop converges more quickly, i.e.,

the $\mathcal{M}_1$ performs best, its $\mathcal{M}_2$'s performance is lower than that of both AdamW and AdaGrad. This suggests that different optimizers profoundly influence the training trajectory from $\mathcal{M}_0$ to $\mathcal{M}_1$ and the resulting $\Delta\boldsymbol{\theta}$. In particular, AdamW, as the optimizer widely used in modern LLM training, achieves an excellent balance between convergence speed (corresponding to $\mathcal{M}_1$'s performance) and optimization direction ($\Delta\boldsymbol{\theta}$, which directly impacts $\mathcal{M}_2$'s performance).

# 4 EXTENDED APPLICATIONS OF EXPO

## 4.1 APPLYING EXPO TO MORE EXISTING LLMS

In § 3.2, we observed that ExPO brings moderate performance improvements to `zephyr-7b-dpo`. This inspires us to apply ExPO to more existing, already-aligned LLMs. We then select twelve open-source models from HuggingFace[6]: (1) Five models trained via **offline DPO**, including `zephyr-7b-alpha/beta` (Tunstall et al., 2023) and `tulu2-7/13/70b` (Ivison et al., 2023); (2) Two models trained via **iterative DPO**, including `snorkel-7b-iter` (Tran et al., 2023) and `llama3-8b-iter` (Dong et al., 2024); (3) Five models trained via **online RLHF**, including `starling-7b-alpha/beta` (Zhu et al., 2023) and `internlm2-1.8/7/20b` (Cai et al., 2024). These models cover a diverse range of model sizes (from 1.8B to 70B) and span three mainstream alignment algorithms widely used in practice.

Based on our hyperparameter search experience for `zephyr-7b-dpo` in § 3.2, for the twelve models above, we conduct a simple grid search for the optimal $\alpha$, using the interval of 0.1 within the range [0.1, 0.5]. In addition to AlpacaEval 2.0, we also evaluate these models on **MT-Bench** (Zheng et al., 2023b), another leading benchmark for assessing instruction-tuned LLMs' general and multi-turn ability. It contains a set of challenging multi-turn open-ended questions covering topics such as writing, role-playing, math, coding, and more. The model-generated answers are judged by GPT-4 via a scalar score (from 1 to 10).

Table 5: Evaluation results on AlpacaEval 2.0 (win rate and LC win rate) and MT-Bench of applying ExPO to existing DPO/RLHF LLMs.

| | Original ($\mathcal{M}_1$) | | | + ExPO ($\mathcal{M}_2$) | | |
|---|---|---|---|---|---|---|
| | **WR** | **LC WR** | **MT-B** | **Win Rate** | **LC Win Rate** | **MT-Bench** |
| $\mathcal{M}_1$ is trained via *Offline DPO* | | | | | | |
| `zephyr-7b-alpha` | 6.7% | 10.0% | 6.85 | **10.6%** (+3.8%) | **13.6%** (+3.6%) | **6.87** (+0.02) |
| `zephyr-7b-beta` | 10.2% | 13.2% | 7.02 | **11.1%** (+0.9%) | **14.0%** (+0.8%) | **7.06** (+0.04) |
| `tulu2-7b` | 8.5% | 10.2% | 6.35 | **11.5%** (+3.0%) | **11.7%** (+1.5%) | **6.38** (+0.03) |
| `tulu2-13b` | 11.2% | 15.5% | 7.00 | **15.6%** (+4.3%) | **17.6%** (+2.1%) | **7.26** (+0.26) |
| `tulu2-70b` | 15.4% | 21.2% | 7.79 | **23.0%** (+7.6%) | **25.7%** (+4.5%) | **8.03** (+0.24) |
| $\mathcal{M}_1$ is trained via *Iterative DPO* | | | | | | |
| `snorkel-7b-iter` | 24.7% | 24.0% | 7.63 | **28.8%** (+4.1%) | **26.4%** (+2.4%) | **7.69** (+0.07) |
| `llama3-8b-iter` | 29.2% | 36.0% | 8.08 | **32.7%** (+3.5%) | **37.8%** (+1.8%) | **8.45** (+0.37) |
| $\mathcal{M}_1$ is trained via *Online RLHF* | | | | | | |
| `starling-7b-alpha` | 15.0% | 18.3% | 7.82 | **18.2%** (+3.2%) | **19.5%** (+1.2%) | **7.91** (+0.09) |
| `starling-7b-beta` | 26.6% | 25.8% | 8.10 | **29.6%** (+3.0%) | **26.4%** (+0.7%) | **8.18** (+0.08) |
| `internlm2-1.8b` | 3.8% | 4.0% | 5.17 | **5.2%** (+1.5%) | **4.3%** (+0.3%) | **5.26** (+0.08) |
| `internlm2-7b` | 20.5% | 18.3% | 7.72 | **28.1%** (+7.6%) | **22.7%** (+4.4%) | **7.80** (+0.08) |
| `internlm2-20b` | 36.1% | 24.9% | 8.13 | **46.2%** (+10.1%) | **27.2%** (+2.4%) | **8.26** (+0.13) |

Table 5 shows that ExPO consistently improves the evaluated LLMs, with notable improvements of up to 10.1% win rate and 4.5% LC win rate on AlpacaEval 2.0 (for `internlm2-20b` and `tulu2-70b`, respectively) and 0.37 on MT-Bench (for `llama3-8b-iter`). This suggests that these already-aligned LLMs may still not have been trained to their optimal status, even though achieving

---

[6]However, we found that many well-known LLMs, such as LLaMA (Touvron et al., 2023b; Dubey et al., 2024) and Qwen (Bai et al., 2023; Yang et al., 2024), only release the final DPO/RLHF checkpoints without the SFT ones. Therefore, we are unable to experiment with these more representative models at this time.

optimal training is inherently challenging given the complexity of hyperparameter tuning. For example, for the `tulu2-7b/13b/70b` models that adopt *identical* training configurations, it is intuitive that larger models, due to their greater capacity, are more difficult to train to optimality, which may also explain why the improvements brought by ExPO increase with their model sizes (e.g., 1.5%, 2.1%, and 4.5% LC win rate for 7B, 13B, and 70B, respectively). ExPO, on the other hand, offers a practical and efficient means to compensate for potential inadequate training of existing LLMs, as it only requires inference-level hardware resources and bypasses the costly training overhead.

## 4.2 Applying ExPO to More Alignment Algorithms

So far, we have primarily applied ExPO to models trained via the dominant DPO or RLHF algorithms (§ 3 and 4.1). However, the formalization and the proposed interpolation hypothesis in § 2 are not tied to any specific alignment algorithm, so we expect that ExPO can also be applied to models trained via other algorithms than DPO or RLHF. To this end, we use a series of Mistral/LLaMA-3 models[7] released by Meng et al. (2024), which are trained via various alignment algorithms and are all initialized from the same SFT checkpoints. These algorithms include: **RRHF** (Yuan et al., 2023), **SLiC-HF** (Zhao et al., 2023a), **IPO** (Azar et al., 2024), **CPO** (Xu et al., 2024), **KTO** (Ethayarajh et al., 2024), **R-DPO** (Park et al., 2024), and **SimPO** (Meng et al., 2024). We refer readers to Meng et al. (2024) for elaboration on these algorithms' optimization objectives as well as the models' training configurations. Following the previous experience, we search the optimal $\alpha$ value within the range of [0.1, 0.5] with the interval of 0.1.

Table 6: Evaluation results on UntraFeedback of applying ExPO to models trained via different alignment algorithms.

| Algorithm | $\mathcal{M}_0$ is SFTed from Mistral | | | $\mathcal{M}_0$ is SFTed from LLaMA-3 | | |
|---|---|---|---|---|---|---|
| | Original ($\mathcal{M}_1$) | + ExPO ($\mathcal{M}_2$) | | Original ($\mathcal{M}_1$) | + ExPO ($\mathcal{M}_2$) | |
| | Reward | $\alpha$ | Reward | Reward | $\alpha$ | Reward |
| SFT ($\mathcal{M}_0$) | 2.97 | - | - | 1.93 | - | - |
| RRHF | 4.71 | 0.1 | 4.73 (+0.02) | 3.02 | 0.5 | 3.15 (+0.13) |
| SLiC-HF | 4.90 | 0.4 | 5.16 (+0.26) | 4.06 | 0.5 | 4.68 (+0.62) |
| IPO | 4.97 | 0.5 | 5.44 (+0.47) | 4.75 | 0.3 | 4.86 (+0.11) |
| CPO | 4.86 | 0.3 | 5.01 (+0.15) | 4.04 | 0.5 | 4.75 (+0.71) |
| KTO | 3.84 | N/A | N/A | 4.48 | 0.4 | 4.67 (+0.19) |
| R-DPO | 5.53 | 0.3 | 5.73 (+0.20) | 4.25 | 0.5 | 4.64 (+0.39) |
| SimPO | 5.88 | 0.1 | 5.95 (+0.07) | 4.89 | 0.4 | 5.21 (+0.32) |

As shown in Table 6, ExPO can be effectively combined with various alignment algorithms, even though these models ($\mathcal{M}_1$) have been *fully trained* according to Meng et al. (2024). Although the only exception is the KTO model on the left, which does not get improved after applying ExPO, we believe this is related to the specific training details of this model, e.g., the configurations leading to overfitting. Overall, we remain optimistic about ExPO's general compatibility with various alignment algorithms.

## 4.3 Discussion on Model Choice

In the interpolation hypothesis in § 2 and in our experiments so far, $\mathcal{M}_0$ is an SFT model and $\mathcal{M}_1$ is one that further undergoes alignment training. A natural question is whether ExPO's interpolation hypothesis can be extended to other model types, for example, where $\mathcal{M}_0$ is a pre-trained model and $\mathcal{M}_1$ is an SFT one. However, based on our attempts with open-source models, we find that in this case, model extrapolation typically fails to improve alignment performance and may even lead to model collapse (e.g., the extrapolated model struggles to generate the EOS token or mistakenly generate special tokens). We speculate that there are two reasons for this phenomenon. **First**, the training from a pre-trained model to an SFT model typically employs a larger learning rate than the subsequent alignment training. The resulting larger $|\Delta\theta|$ could invalidate ExPO's underlying

---

[7]`https://huggingface.co/princeton-nlp?search_models=+Mistral-7B-Base-SFT`
`https://huggingface.co/princeton-nlp?search_models=+Llama-3-Base-8B-SFT`

first-order approximation (§ 3.3). **Second**, one important function of SFT is to adapt the model to the instruction/chat-style input template (Zheng, 2024). Extrapolation will amplify this part of the information in $\Delta\theta$, which could thus lead to model collapse and impact normal response generation.

## 5 RELATED WORK

**LLM Alignment** Modern large language models (LLMs) are first pre-trained on massive textual corpora with the unsupervised language modeling objective (Brown et al., 2020; Touvron et al., 2023b; Dubey et al., 2024), and then fine-tuned to learn to follow human instructions (OpenAI, 2022; 2023; Ji et al., 2023). The current fine-tuning paradigm typically contains two steps: supervised fine-tuning (SFT) and *human preference optimization*. Our work focuses on the later step, which aims to adjust the model's response distribution to better *align with human preferences*. In this process, the model is usually trained on preference data ("A is better than B"; Zhao et al. 2023b; Zheng et al. 2023a), thus learning to assign higher probabilities to human-preferred responses over the disfavored ones. Common implementations for human preference optimization include Reinforcement Learning from Human Feedback (RLHF; Ouyang et al. 2022; Schulman et al. 2017), Direct Preference Optimization (DPO; Rafailov et al. 2023), and many other DPO's variants or competitors (Azar et al., 2024; Xu et al., 2024; Ethayarajh et al., 2024; Park et al., 2024; Meng et al., 2024). Given LLMs' gigantic parameters, the processes from pre-training to SFT and the alignment training still require expensive computational resources. Therefore, exploring more efficient alignment methods to reduce training overhead has always been an important and compelling research challenge (Ji et al., 2024a). To address this challenge, we propose the ExPO method, which has demonstrated promising efficacy in expediting LLM alignment.

There is another line of work that attempts to bypass the expensive alignment training by blending multiple models' token predictions during the inference time (Liu et al., 2021; Lu et al., 2024; Liu et al., 2024), usually referred to as *inference-time alignment methods*. In comparison to ExPO, these inference-time methods often require more complex and varied implementations of model inference, which are not typically supported by existing high-performance LLM inference infrastructures (e.g., vLLM). This inconvenience not only reduces the practical efficiency of model inference but also significantly increases the cost of their hyperparameter search processes. In contrast, ExPO only involves regular inference of a single model, which can be seamlessly supported by existing infrastructures, thereby inheriting the merit in inference efficiency.

**Model Interpolation** Model interpolation (or model averaging) is a commonly used technique in machine learning. It typically involves training multiple models with different random initializations or data subsets and then interpolating the weights of these models to obtain a new model with stronger out-of-distribution generalization (Izmailov et al., 2018; Lin et al., 2024; Wortsman et al., 2022; Lin et al., 2023). This technique is based on the (linear) mode connectivity of neural networks (Garipov et al., 2018; Entezari et al., 2022; Zhao et al., 2020; Frankle et al., 2020). Specifically, prior work found that multiple local optima in the parameter space can often be connected by low-loss (linear) paths, particularly for models with residual connection structures (He et al., 2016). This may explain why model interpolation can produce new, functional models when applied to LLMs (as we observed in § 1), as residual connection has become a dominant choice of architecture design in modern LLMs like LLaMA (Touvron et al., 2023a). We notice that recent LLMs have widely adopted model interpolation, as exemplified by Gemma-2 (Gemma et al., 2024) and LLaMA-3 (Dubey et al., 2024), possibly also for further enhancement in out-of-distribution generalization.

Unlike interpolating between multiple fine-tuned $\mathcal{M}_1$, our work begins with the attempt to interpolate between models before and after alignment training ($\mathcal{M}_0$ and $\mathcal{M}_1$). The resulting observation in § 1 inspires the hypothesis and the proposal of the ExPO (model extrapolation) method.

## 6 CONCLUSION

We demonstrate the efficacy of the ExPO (model extrapolation) method in enabling more efficient LLM alignment with human preferences. ExPO is based on the hypothesis that, for a partially-trained model $\mathcal{M}_1$, we may obtain a better-aligned model $\mathcal{M}_2$ by simply extrapolating the model weights along the weight difference $\Delta\theta$ between $\mathcal{M}_1$ and its initial SFT checkpoint $\mathcal{M}_0$, thus

bypassing further training of $\mathcal{M}_1$ and directly reaching better alignment performance. We empirically validate this hypothesis through controlled experiments, where we show that the DPO model trained with 20% steps can be boosted to outperform the fully-trained one. Furthermore, we extend ExPO's application to twelve existing, already-aligned LLMs, showing that ExPO consistently improves their performance on the mainstream LLM benchmarks AlpacaEval 2.0 and MT-Bench. This suggests that ExPO can also serve as a practical and efficient means to compensate for potential inadequate alignment training of existing LLMs. Overall, our work highlights the utility of model extrapolation in efficient LLM alignment, which may inspire future research in this direction.

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

# A  LIMITATIONS

**Theoretical Analysis**  Although we provide a preliminary discussion for why ExPO can work (§ 3.3), a more rigorous theoretical analysis is necessary to fully understand the factors contributing to its effectiveness. Future work could establish more profound theoretical foundations for the underlying mechanisms of ExPO (model extrapolation).

**Hyperparameter Search**  The current ExPO adopts the simplest form of uniform extrapolation (§ 2) and requires manual hyperparameter search for $\alpha$. Future work could explore how to determine the optimal $\alpha$ automatically and adaptively (i.e., using different $\alpha$ values for different model modules). For example, the information from optimizer states and parameter gradients during the later stage of alignment training could be useful for this purpose.

**Alignment Tax**  While ExPO makes notable improvements in instruction-following ability and alignment with human preferences, this seems not "free" and may instead incur an additional *alignment tax*, a widely-observed issue in human preference optimization algorithms (Ouyang et al., 2022; Dong et al., 2024; Meng et al., 2024), which indicates the possible fluctuations or drops in downstream task performance after human preference optimization. We evaluate the models in § 3.2 and 4.1 on the six downstream tasks (Clark et al., 2018; Zellers et al., 2019; Hendrycks et al., 2021; Lin et al., 2022; Sakaguchi et al., 2021; Cobbe et al., 2021) from the Open LLM Leaderboard[8] (v1; Beeching et al. 2023). We find that in most cases, ExPO amplifies the alignment tax introduced by the alignment training (from $\mathcal{M}_0$ to $\mathcal{M}_1$). For example, for the partially-trained models in § 3.2 (Figure 3), the original DPO models ($\mathcal{M}_1$) show improvements over the initial SFT model ($\mathcal{M}_0$) on TruthfulQA and declines on GSM8K, while applying ExPO ($\mathcal{M}_2$) leads to further improvements or declines, respectively. For the existing, fully-trained LLMs in § 4.1, the amplification of the alignment tax by ExPO is usually smaller as shown in Figure 4, suggesting a trade-off between the alignment training overhead (from $\mathcal{M}_0$ to $\mathcal{M}_1$) and the additional alignment tax brought by ExPO (from $\mathcal{M}_1$ to $\mathcal{M}_2$).

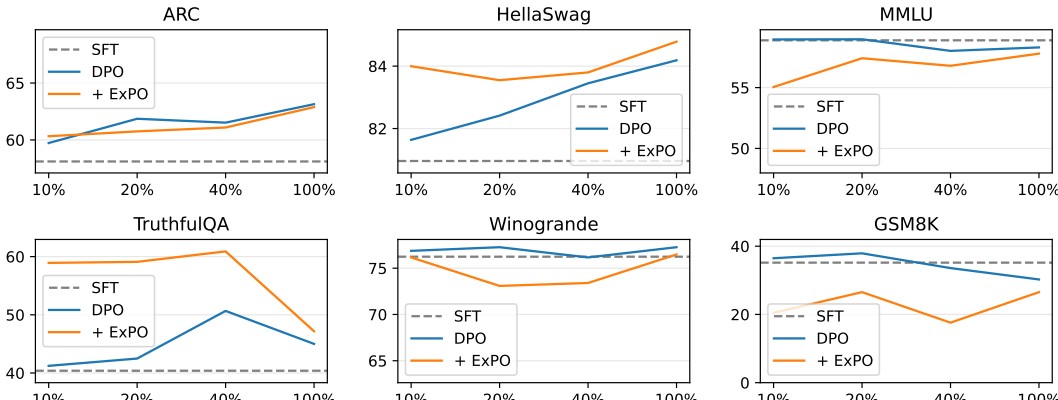

Figure 3: Evaluation results for the models in § 3.2 on downstream tasks. The x-axis denotes the proportions of training steps. As the "cost" of simply improving instruction-following ability and alignment with human preferences, ExPO can also amplify the alignment tax introduced by the alignment training.

---

[8]We employ the evaluation implementation of Eleuther's lm-evaluation-harness (`https://github.com/EleutherAI/lm-evaluation-harness`, version 0.4.4). Note that the mismatch of input templates used for chat-style evaluations (e.g., AlpacaEval 2.0 and MT-Bench) and for these downstream task evaluations could also contribute to the observed alignment tax, as discussed in Meng et al. (2024).

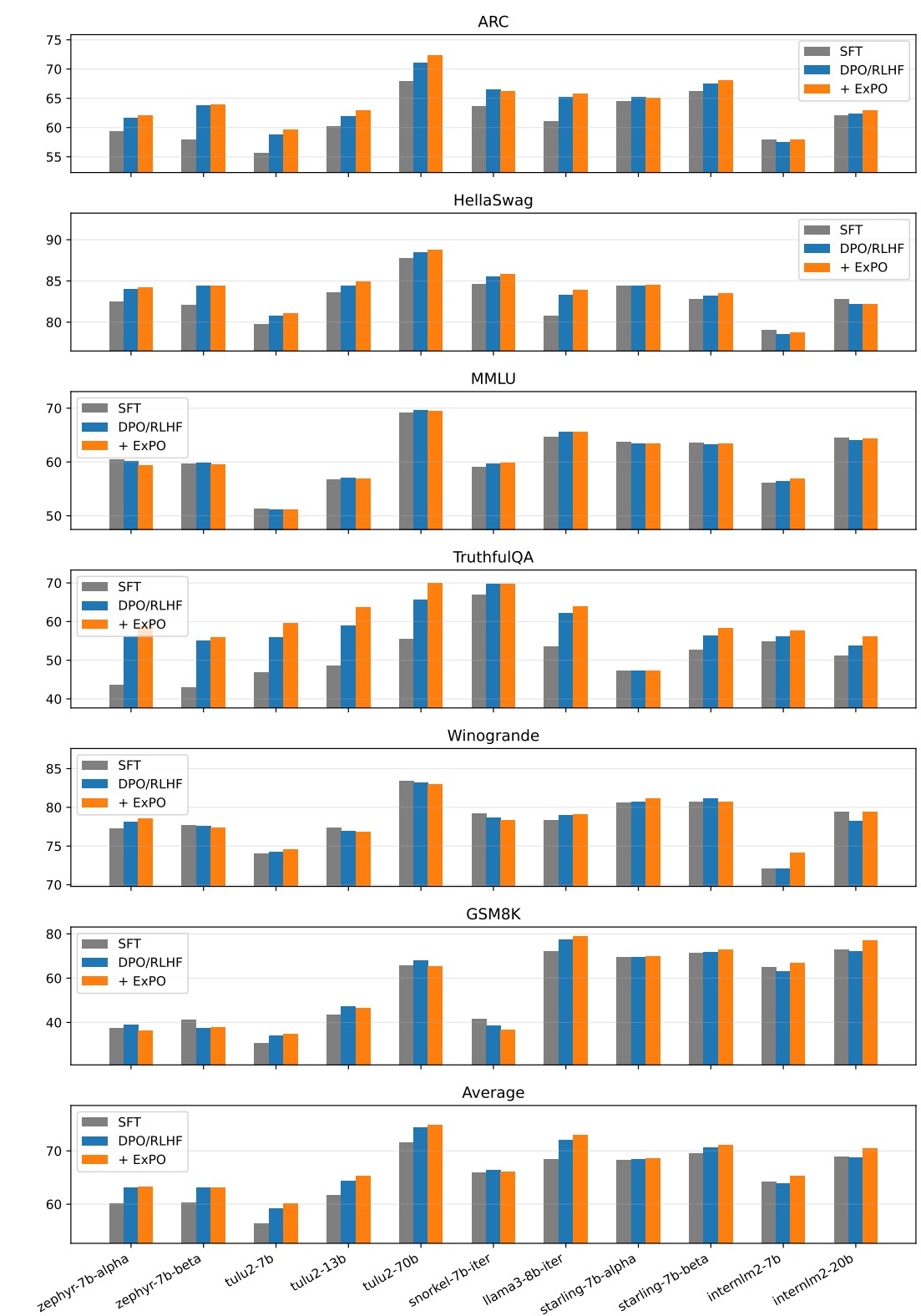

Figure 4: Evaluation results for the LLMs in § 4.1 on downstream tasks. For these fully-trained models, the additional alignment tax brought by ExPO is usually smaller, suggesting a trade-off between the alignment training overhead (from $\mathcal{M}_0$ to $\mathcal{M}_1$) and the additional alignment tax brought by ExPO (from $\mathcal{M}_1$ to $\mathcal{M}_2$).

## B  HYPERPARAMETER SEARCH DETAILS

Table 7: Hyperparameter search results for $\alpha$ in § 3.2 and 4.1.

|  | Search Interval | Optimal $\alpha$ |
| --- | --- | --- |
| **Models in § 3.2** (binary/grid search) | | |
| DPO (10% data) | 1.0 | 8.0 |
| DPO (20% data) | 0.5 | 2.5 |
| DPO (40% data) | 0.1 | 0.5 |
| zephyr-7b-dpo | 0.1 | 0.3 |
| **Models in § 4.1** (grid search within [0.1, 0.5]) | | |
| zephyr-7b-alpha | 0.1 | 0.3 |
| zephyr-7b-beta | 0.1 | 0.1 |
| tulu2-7/13/70b | 0.1 | 0.5 |
| snorkel-7b-iter | 0.1 | 0.3 |
| llama3-8b-iter | 0.1 | 0.3 |
| starling-7b-alpha | 0.1 | 0.2 |
| starling-7b-beta | 0.1 | 0.5 |
| internlm2-1.8/7/20b | 0.1 | 0.5 |

We present the hyperparameter search results in Table 7. For the partially-trained DPO models in § 3.2, we plot in Figure 5 the reward distribution on the UltraFeedback development set (1K instructions). We also plot in Figure 6 how $\mathcal{M}_2$'s expected reward score and response length vary with the different $\alpha$ values.

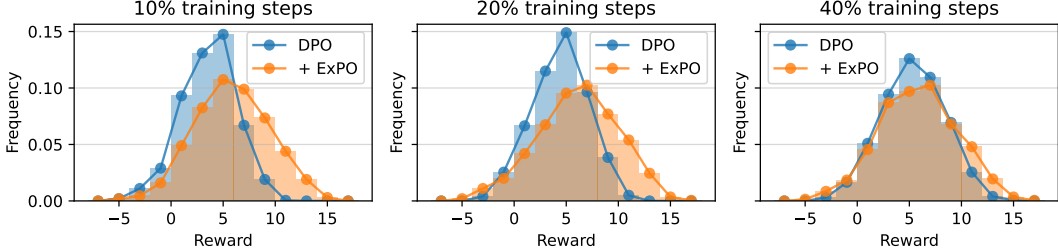

Figure 5: Reward distribution on the UltraFeedback development set (1K instructions) for the partially-trained DPO models in § 3.2.

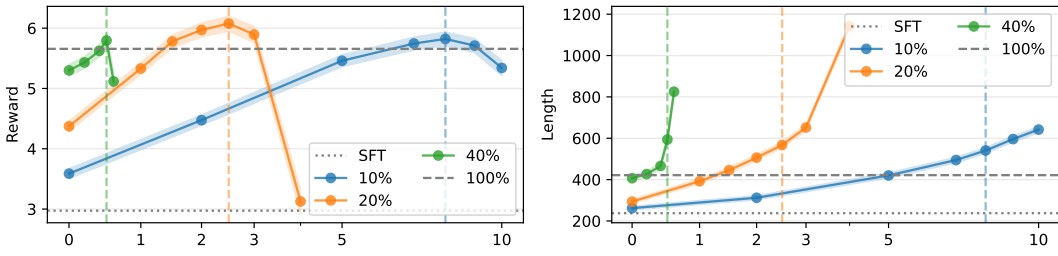

Figure 6: $\mathcal{M}_2$'s expected reward scores and response lengths on UltraFeedback varying with $\alpha$ (x-axis) for the partially-trained DPO models in § 3.2. Dashed vertical lines correspond to the optimal $\alpha$ values. $\alpha = 0$ indicates that EXPO is not applied (i.e., $\mathcal{M}_1$).

## C  INFERENCE DETAILS

For model inference in the experiments, we employ the vLLM (Kwon et al., 2023) library for high-throughput inference. We use top-$k$ ($k = 40$) and nucleus sampling (Holtzman et al., 2020) ($p = 0.9$) with a temperature of 0.7. To avoid repetition in generated texts, we set both the factors of presence penalty and frequency penalty to 0.1. We set the sampling random seed to 42.

# D    HUGGINGFACE MODELS

| | | HuggingFace Model ID |
|---|---|---|
| **Reward model** | | `weqweasdas/RM-Mistral-7B` |
| **Models in § 3** | | |
| `zephyr-7b-dpo` | Pre-trained | `mistralai/Mistral-7B-v0.1` |
| | $\mathcal{M}_0$ | `alignment-handbook/zephyr-7b-sft-full` |
| | $\mathcal{M}_1$ | `alignment-handbook/zephyr-7b-dpo-full` |
| **Models in § 4.1** | | |
| `zephyr-7b-alpha` | $\mathcal{M}_0$ | `HuggingFaceH4/mistral-7b-sft-alpha` |
| | $\mathcal{M}_1$ | `HuggingFaceH4/zephyr-7b-alpha` |
| `zephyr-7b-beta` | $\mathcal{M}_0$ | `HuggingFaceH4/mistral-7b-sft-beta` |
| | $\mathcal{M}_1$ | `HuggingFaceH4/zephyr-7b-beta` |
| `tulu2-7b` | $\mathcal{M}_0$ | `allenai/tulu-2-7b` |
| | $\mathcal{M}_1$ | `allenai/tulu-2-dpo-7b` |
| `tulu2-13b` | $\mathcal{M}_0$ | `allenai/tulu-2-13b` |
| | $\mathcal{M}_1$ | `allenai/tulu-2-dpo-13b` |
| `tulu2-70b` | $\mathcal{M}_0$ | `allenai/tulu-2-70b` |
| | $\mathcal{M}_1$ | `allenai/tulu-2-dpo-70b` |
| `snorkel-7b-iter` | $\mathcal{M}_0$ | `mistralai/Mistral-7B-Instruct-v0.2` |
| | $\mathcal{M}_1$ | `snorkelai/Snorkel-Mistral-PairRM-DPO` |
| `llama3-8b-iter` | $\mathcal{M}_0$ | `RLHFlow/LLaMA3-SFT` |
| | $\mathcal{M}_1$ | `RLHFlow/LLaMA3-iterative-DPO-final` |
| `starling-7b-alpha` | $\mathcal{M}_0$ | `openchat/openchat_3.5` |
| | $\mathcal{M}_1$ | `berkeley-nest/Starling-LM-7B-alpha` |
| `starling-7b-beta` | $\mathcal{M}_0$ | `openchat/openchat-3.5-0106` |
| | $\mathcal{M}_1$ | `Nexusflow/Starling-LM-7B-beta` |
| `internlm2-1.8b` | $\mathcal{M}_0$ | `internlm/internlm2-chat-1_8b-sft` |
| | $\mathcal{M}_1$ | `internlm/internlm2-chat-1_8b` |
| `internlm2-7b` | $\mathcal{M}_0$ | `internlm/internlm2-chat-7b-sft` |
| | $\mathcal{M}_1$ | `internlm/internlm2-chat-7b` |
| `internlm2-20b` | $\mathcal{M}_0$ | `internlm/internlm2-chat-20b-sft` |
| | $\mathcal{M}_1$ | `internlm/internlm2-chat-20b` |
| **Mistral-based Models in § 4.2** | | |
| SFT | $\mathcal{M}_0$ | `alignment-handbook/zephyr-7b-sft-full` |
| RRHF | $\mathcal{M}_1$ | `princeton-nlp/Mistral-7B-Base-SFT-RRHF` |
| SLiC-HF | $\mathcal{M}_1$ | `princeton-nlp/Mistral-7B-Base-SFT-SLiC-HF` |
| IPO | $\mathcal{M}_1$ | `princeton-nlp/Mistral-7B-Base-SFT-IPO` |
| CPO | $\mathcal{M}_1$ | `princeton-nlp/Mistral-7B-Base-SFT-CPO` |
| KTO | $\mathcal{M}_1$ | `princeton-nlp/Mistral-7B-Base-SFT-KTO` |
| R-DPO | $\mathcal{M}_1$ | `princeton-nlp/Mistral-7B-Base-SFT-RDPO` |
| SimPO | $\mathcal{M}_1$ | `princeton-nlp/Mistral-7B-Base-SFT-SimPO` |
| **LLaMA-3-based Models in § 4.2** | | |
| SFT | $\mathcal{M}_0$ | `princeton-nlp/Llama-3-Base-8B-SFT` |
| RRHF | $\mathcal{M}_1$ | `princeton-nlp/Llama-3-Base-8B-SFT-RRHF` |
| SLiC-HF | $\mathcal{M}_1$ | `princeton-nlp/Llama-3-Base-8B-SFT-SLiC-HF` |
| IPO | $\mathcal{M}_1$ | `princeton-nlp/Llama-3-Base-8B-SFT-IPO` |
| CPO | $\mathcal{M}_1$ | `princeton-nlp/Llama-3-Base-8B-SFT-CPO` |
| KTO | $\mathcal{M}_1$ | `princeton-nlp/Llama-3-Base-8B-SFT-KTO` |
| R-DPO | $\mathcal{M}_1$ | `princeton-nlp/Llama-3-Base-8B-SFT-RDPO` |
| SimPO | $\mathcal{M}_1$ | `princeton-nlp/Llama-3-Base-8B-SFT-SimPO` |

