# OpenReview forum: "Model Extrapolation Expedites Alignment"
_ICLR.cc/2025/Conference — ICLR 2025 Conference Withdrawn Submission_

### Official Review · Reviewer_4oMY · 2024-10-21

**Soundness:** 2
**Presentation:** 2
**Contribution:** 2
**Rating:** 3
**Confidence:** 4

**Summary:**

The paper introduces **EXPO** (model extrapolation), a method to improve the efficiency of aligning large language models (LLMs) with human preferences. Instead of fully training models, EXPO extrapolates model weights from a partially-trained model, significantly reducing training time while improving performance. Through experiments, the method shows that models trained with fewer steps can outperform fully-trained ones, validated on benchmarks like AlpacaEval 2.0 and MT-Bench. EXPO is applicable to a wide range of LLMs, making alignment more efficient and less computationally expensive.

**Strengths:**

1. The paper tackles an important problem—reducing the computational cost of aligning large language models—which is crucial for scaling models efficiently.

2. The explanation of the method is clear and easy to follow, with helpful figures that enhance understanding.

3. The authors show EXPO’s ability to cut down alignment training costs, which supports their claim.

**Weaknesses:**

1. **Limited theoretical foundation**: The paper presents an interesting empirical finding with E X PO, but lacks a rigorous theoretical analysis of why it works. The discussion in Section 3.3 on why E X PO can work is somewhat speculative. A more formal theoretical treatment, perhaps drawing connections to optimization theory or analyzing the loss landscape, would strengthen the paper's contribution.

2. **Limited analysis of failure cases**: While some negative results are reported (e.g. for KTO algorithm in Table 6), there's little in-depth analysis of when and why E X PO fails. A more comprehensive error analysis would provide insights into the method's limitations.

3. **Limited analysis of extrapolation's impact on model calibration**: While the paper focuses on performance metrics, it doesn't investigate how E X PO affects the model's calibration. An analysis of how extrapolation impacts the model's uncertainty estimates and confidence calibration would be valuable, especially for safety considerations.

4. **Insufficient investigation of extrapolation instability**: The authors mention in Section 4.3 that extrapolating from pre-trained to SFT models can lead to model collapse, but they don't provide a rigorous analysis of this phenomenon.

5. **Lack of analysis on the impact of different λ values in DPO**: In Table 4, the authors test λ values of 0.001, 0.01, and 0.1, but don't provide a detailed analysis of how these different λ values affect the optimization landscape and consequently E X PO's performance. A more granular sweep of λ values and an analysis of how they interact with the optimal α could provide insights into the method's sensitivity to DPO hyperparameters.

**Questions:**

I encourage the authors to address the points raised in the weaknesses section and to conduct additional experiments where further investigation is required.

---

### Official Review · Reviewer_kY78 · 2024-10-23

**Soundness:** 1
**Presentation:** 4
**Contribution:** 2
**Rating:** 3
**Confidence:** 4

**Summary:**

The paper presents a method called EXPO for improving the accuracy of language models fine-tuned with DPO/RLHF. The intuition is that model interpolation can produce a model with accuracy in between the original model and the reward-fine-tuned model. So, model extrapolation should be able to produce a model with greater accuracy. By partially training a model, then performing extrapolation, a more accurate final model can be obtained with less compute. The method searches along the line defined by M0 (the initialization for reward-fine-tuning (e.g. the SFT model)) and M1 (the partially-reward-fine-tuned model) to find a more accurate model. The paper presents results for a variety of open-source models.

**Strengths:**

- The paper studies a method for improving reward-fine-tuned models. This is an important problem in current LLM research.
- The paper is well-written and easy to understand. Figure 2 presents a clear indication of how the method works. Section 2 clearly presents the hypothesis, and the method is neatly summarized by equation 2.
- The results presented are significant. Table 1 demonstrates significant gains in Win Rate using the proposed method. Table 5 demonstrates that the results extend to other models. Table 6 demonstrates the algorithm extends to other alignment methods.
- The paper presents ablations to help understand the method. The ablation in Table 3 (extending the training regime or increasing the learning rate) is particularly important for understanding why the method differs in performance from simply extending training.

**Weaknesses:**

1) To me, there seems to be a disconnect between the theory presented in Section 2 and the results presented. The results seem to point towards different conclusions that contradict the theory.
  a) Line 192-193: Extrapolation strongly improves the results obtained by "DPO 100%". This is not predicted by the theory presented. The theory presented essentially states, "we can partially train an RLHF model, then predict the result of fully training. The resulting predicted model will achieve the accuracy that would have been obtained by full training". Instead, the results demonstrate that the extrapolation method results in strong gains even when M1 is a fully trained model. (This seems at first like a sign that M1 is undertrained, but this theory is rebuked in Table 3).
  b) Based on the theory, the extrapolated model M2 should be most accurate if the search for M2 occurs along the line between M0 and the "fully trained" M1. This line is approximated by training a proxy (e.g. $M_1 ^{20\%}$) and searching along the line between M0 and the proxy. But for some reason, this proxy yields *better* results than searching the line between M0 and M1 (22.7% win rate compared to 18.0% win rate). Thus, a misalignment of the search space is beneficial to the model. This seems like a clear sign that something not predicted by the theory is causing the benefits in accuracy.
  c) Given the above observations, it seems like the true gains in accuracy could be coming from the fact that ExPO models are selected using a method that finds the best-performing model on the UltraFeedback dev set as calculated by another reward model. If so, this is an interesting discovery in itself, but it's not what the paper focuses on (and it's just a theory that could be completely wrong). My point here is, it seems like the theory presented doesn't match the given results, and there's some other reason for the gains in accuracy.

Line 225-229: The authors are trying to show that $|\alpha \delta \theta|$ is small. They argue that $\alpha$ decreases through training, then conclude that $|\alpha \delta \theta|$ is small. But they do not discuss the fact that $\delta \theta$ is (presumably) increasing during training. This justification seems incomplete/incorrect to me.

Section 3.4: The authors successfully demonstrate in this section that length bias can be an issue for the method. But this does not prove that length bias is the only issue causing reduced accuracy when training for longer. Also, shouldn't the same logic (length bias being an issue) apply to the standard training procedure? Yet, training for the full recipe produces optimal results, as opposed to training for a fraction of training steps. Why is length bias only an issue for ExPO?

**Questions:**

The main issue that I'm looking to see resolved is in (1) above. The results of the paper are good, but there seems to be a mismatch between the theory and the method. It leaves me wondering if there's a different reason that the method works. See discussion in (1).

---

> ### Author Response · Authors · 2024-11-15
> **Rebuttal by Authors (1/2)**
>
> Thank you for your review and feedback. We address your concerns and questions as follows.
>
> For your main issue, we first clarify:
>
> > The theory presented essentially states, "we can partially train an RLHF model, then predict the result of fully training. The resulting predicted model will achieve the accuracy that would have been obtained by full training".
> >
> > (We first clarify that our term "full training" refers to the baseline that uses the complete training steps. This does not imply that the model is trained to optimality.)
>
> Our formalization in Section 2 does not imply that ExPO is "predicting full training". ExPO hypothesizes that: for a partially-trained model $M_1$, there may exist a better-performing model $M_2$ on the extrapolation trajectory from $M_0$ to $M_1$. This $M_2$ **may not (and very likely cannot)** be obtained by continuing to train $M_1$, given the inherent uncertainty and non-uniqueness of neural network optimization and convergence (for large language models, this also involves learning rate dynamics, gradient clipping, etc.). **ExPO does not aim to, nor can it "predict" the results of full training.**
>
> > ... the results demonstrate that the extrapolation method results in strong gains even when M1 is a fully trained model. (This seems at first like a sign that M1 is undertrained, ...)
>
> As you noted, we conjecture this is because $M_1$ has not been trained to optimality. Given the inherent challenges in training LLMs, alignment training is very likely to be suboptimal (as also evidenced in Section 4.1).
>
> > ... Based on the theory, the extrapolated model M2 should be most accurate if the search for M2 occurs along the line between M0 and the "fully trained" M1.
>
> We respectfully disagree. A fully-trained model (i.e., the baseline using complete training steps) is not necessarily trained to optimality, thus its indicated $\Delta\theta$ may not be optimal (e.g., it may overfit or capture spurious features in the data, as stated in lines 240-242). A partially-trained model may indicate a more accurate $\Delta\theta$, in which case the optimal model on this extrapolation trajectory would outperform the optimal model on the trajectory indicated by the fully-trained model (according to our discussion in lines 266-268, since $\theta_2=\theta_0+(1+\alpha)\Delta\theta$, the achievable performance of $M_2$ is uniquely determined by $\Delta\theta$).
>
> > This line is approximated by training a proxy (e.g. $M_1^{20\\%}$) and searching along the line between M0 and the proxy.
>
> Based on the previous responses, ExPO is not "predicting full training", thus the "proxy" is not approximating the trajectory from $M_0$ to the fully-trained $M_1$.
>
> > But for some reason, this proxy yields *better* results than searching the line between M0 and M1 (22.7% win rate compared to 18.0% win rate). Thus, a misalignment of the search space is beneficial to the model. This seems like a clear sign that something not predicted by the theory is causing the benefits in accuracy.
>
> Based on the previous responses, ExPO is not "predicting full training", and the "proxy" does not necessarily (and likely will not) indicate the same extrapolation direction $\Delta\theta$ as the fully-trained model. Therefore, the experimental results do not contradict Section 2. On the other hand, the performance improvements come from the accuracy of $\Delta\theta$, which is exactly what we discuss and analyze in detail in Section 3.3-3.5.
>
> > Given the above observations, it seems like the true gains in accuracy could be coming from the fact that ExPO models are selected using a method that finds the best-performing model on the UltraFeedback dev set as calculated by another reward model.
>
> We respectfully disagree. As stated in lines 266-268, since $\theta_2=\theta_0+(1+\alpha)\Delta\theta$, and $M_0$ is fixed, the achievable performance of $M_2$ is uniquely determined by $\Delta\theta$. The development set and reward model only assist in searching for $\alpha$, they do not change $\Delta\theta$, and thus naturally cannot change the achievable performance of $M_2$ (although due to limitations of the reward model, the searched optimal $\alpha$ may vary slightly; however, improving reward models to enhance search results is beyond the scope of our work).
>
> > My point here is, it seems like the theory presented doesn't match the given results, and there's some other reason for the gains in accuracy.
>
> Based on the above discussion, we respectfully suggest that you may have misunderstood the idea of ExPO (which is not "predicting full training"), and we have provided clarification and explanation. We hope this addresses your concerns about ExPO's soundness.

---

> > ### Comment · Reviewer_kY78 · 2024-11-16
> > **Response to Rebuttal 1/2**
> >
> > Thanks for the thorough discussion. Also, my comment used to be formatted nicely, and it got turned into one huge paragraph when I submitted, sorry about that.
> >
> > Regarding the fact that you're not trying to extrapolate the same optima that the fully trained M1 reaches, but you instead hypothesize the existence of some other (better) M2: Ok sure. Consider wording adjustments in your paper. For example: in line 19 of the abstract, when you say "hypothetical better-aligned model M2", make it clear that you mean better aligned that the *fully trained* M1, not just the partially trained M1 that you allude to on 17.
> >
> > I still have a few points though:
> > 1. Let's call M1 a normal fully trained model, and M0.5 a partially trained model. You hypothesize the existence of an M2 that outperforms M1, and the M2 lies along the line between M1 and M0.5. So then, your method searches along the 1-dimensional space defined by the line through M0 and M0.5. And this outperforms M2, which is optimized over an extremely high dimensional space. Why would we expect this to happen? And, why wouldn't the normal optimization procedure (used to produce M1) find a better solution (e.g. M2)? The optimization was already heading in the right direction (e.g. the line from M0 to M0.5). Why would it veer off course and find a solution worse than M2?
> >
> >
> > 2. Going back to this point:
> > > ExPO models are selected using a method that finds the best-performing model on the UltraFeedback dev set as calculated by another reward model
> >
> > Your response:
> > > The development set and reward model only assist in searching for $\alpha$, they do not change $\Delta \theta$, and thus naturally cannot change the achievable performance of M2.
> >
> > My point is that they assist in searching for $\alpha$. That means that your solution point M2 is conditioned on the UltraFeedback dev set (through alpha), whereas M1 is not.
> >
> > Imagine you're searching in a region of near-constant loss, for a model that performs well. M1 is trying to find the center of this region, which is a noisy process. And the training process has satured. Extending training for a lot longer doesn't really improve results. By contrast, the method for choosing M2 is, (1) pick a line through the region of near-constant loss, then (2) find the M2 along this line that performs best on the UltraFeedback dev set.
> >
> > Thus, the advantage for M2 is, instead of trying to find the best loss in a near-constant region, it gets to consult the UltraFeedback dev set to find an optima that will perform better on other datasets.
> >
> > This argument doesn't really hinge on the assumption that the loss is nearly constant, I just find it easier to describe that way. The point is, M2 is conditioned on UltraFeedback through $\alpha$. I'm worried that this is the real source of gains.

---

> > > ### Author Response · Authors · 2024-11-16
> > > **Follow-up Response 1/2**
> > >
> > > Thank you for your response. We continue to address your concerns:
> > >
> > > > Point 1
> > >
> > > We do not assume that M2 will outperform (fully-trained) M1. In ExPO's hypothesis, we have M0 and M0.5, and obtain M2 through extrapolation. As long as M2 can outperform M0.5, ExPO's hypothesis can hold — we have no intention or purpose of comparing the obtained M2 with fully-trained M1 (obtained by extending M0.5's training). We simply aim to obtain an M2 that's better than M0.5, which would verify the ExPO hypothesis. We believe the formulation in Section 2 is clear. We acknowledge that the term "partially-trained" in line 17 of the abstract might cause ambiguity. Thanks for your suggestion; we will revise the wording.
> > >
> > > Regarding the finding that M2 outperforms M1, we were also surprised. We conjecture this is because training **doesn't always proceed in the ideally correct direction**: as stated in lines 240-242, longer training might cause the model to capture spurious features in the data (such as length bias), thus reducing the accuracy of $\Delta\theta$. We verified this in Section 3.4 through controlling data quality (specifically, length bias in the data). Ideally, if the data were diverse, sufficient, and free of spurious features and biases, the $\Delta\theta$ given by M1 and M0.5 should not differ significantly. However, this is difficult to achieve in practice. Moreover, dynamic learning rates, gradient clipping, and other implementation details in LLM training further complicate this issue.
> > >
> > > > Point 2
> > >
> > > We understand your concern: M1 doesn't use development set information, while M2 does.
> > >
> > > On one hand, we supplemented results of further training M1 on the development set (still evaluating expected reward scores on the **development set**). As shown, M2 still outperforms the additionally trained M1.
> > >
> > > |            | M1   | M1 trained on dev | M2 (M1 + ExPO) |
> > > | ---------- | ---- | ----------------- | -------------- |
> > > | 10% steps  | 3.59 | 3.73 (+0.14)      | 5.82 (+2.23)   |
> > > | 20% steps  | 4.37 | 4.46 (+0.09)      | 6.08 (+1.71)   |
> > > | 40% steps  | 5.30 | 5.33 (+0.03)      | 5.80 (+0.50)   |
> > > | 100% steps | 5.66 | 5.64 (-0.02)      | 5.81 (+0.15)   |
> > >
> > > On the other hand, we believe your concern touches on the model selection problem faced by all machine learning algorithms: we need to define meaningful development sets and evaluate on them in some way to select appropriate models. In this sense, choosing development sets closer to downstream tasks or actual applications (we chose the UltraFeedback development set in our experiments) and more accurate evaluation methods (we used an open-source reward model) would improve model selection results.
> > >
> > > Finally, as shown in Figure 6 of the appendix, even if we search for slightly different $\alpha$ values using other development sets or reward models (as long as they are reasonable), performance gains still exist, differing only in magnitude. This is because (as previously stated) whether M2 can produce gains is uniquely determined by $\Delta\theta$, while the development set and reward model used to assist in searching for $\alpha$ only affect how much gain we can obtain.

---

> ### Author Response · Authors · 2024-11-15
> **Rebuttal by Authors (2/2)**
>
> > Line 225-229: The authors are trying to show that |αδθ| is small. They argue that α decreases through training, then conclude that |αδθ| is small. But they do not discuss the fact that δθ is (presumably) increasing during training.
>
> We have already discussed this fact in lines 227-228: "This suggests that models trained with more steps, whose $|\Delta\theta|$ is usually larger, require smaller optimal $\alpha$ values."
>
> > Section 3.4: The authors successfully demonstrate in this section that length bias can be an issue for the method. But this does not prove that length bias is the only issue causing reduced accuracy when training for longer. Also, shouldn't the same logic (length bias being an issue) apply to the standard training procedure? Yet, training for the full recipe produces optimal results, as opposed to training for a fraction of training steps. Why is length bias only an issue for ExPO?
>
> We have not claimed or proved that length bias is the "only" issue affecting the accuracy of $\Delta\theta$. We want to point out that **"This issue" in line 242 refers to "increasing the training steps makes the model more prone to learning spurious features from the training data", not length bias**. If this wording caused confusion, we sincerely apologize. We use length bias here as an example of data quality mainly because it is a typical example that is simple and amenable to controlled experiments.
>
> Finally, we hope our responses can address your concerns, and we look forward to your favorable consideration of our work.

---

> > ### Comment · Reviewer_kY78 · 2024-11-15
> > **Response to Rebuttal 2/2**
> >
> > Thanks for your reply, I appreciate the discussion.
> >
> > > We have already discussed this fact in lines 227-228: "This suggests that models trained with more steps, whose $|\Delta \theta|$ is usually larger, require smaller optimal $\alpha$ values."
> >
> > Understood, but I don't see how this proves that $|\alpha \Delta \theta|$ is small.
> >
> > To simplify notation, I'll use $|ab|$. Now imagine I argue that $|ab|$ is small. And I demonstrate that $a$ takes on decreasing values [8, 2.5, 0.5, 0.3], while $b$ takes on increasing values [3, 9.6, 48, 80]. Well, $|ab|$ is the same for all pairs $a_i, b_i$. And the fact that $a$ is decreasing over the sequence doesn't prove anything about $|ab|$ because $b$ is increasing. In another hypothetical example, $b$ could be increasing at an even faster rate than $a$ is decreasing.
> >
> > In other words, your argument doesn't hold if, say, $\Delta \theta$ is increasing faster than $\alpha$ is decreasing.
> >
> > More relevantly, for the Taylor approximation to hold, $|\alpha \Delta \theta|$ needs to be small. So my question is, "small compared to what?" The fact that alpha decreases when $\Delta \theta$ increases doesn't tell me anything about the magnitude $|\alpha \Delta \theta|$ compared to other terms in the taylor approximation.
> >
> > The rough answer to my question of "small compared to what" is "small compared to other relevant terms in the Taylor approximation". But I am not sure what a typical value of $|\alpha \Delta \theta|$ is, since it's a very high-dimensional object. Did you calculate sample values? Can you prove that it's small compared to other terms in the Taylor approximation?

---

> > > ### Author Response · Authors · 2024-11-16
> > > **Follow-up Response 2/2**
> > >
> > > We understand your point. Our discussion about first-order approximation is merely an attempt to provide a preliminary explanation for ExPO's effectiveness, and the condition for first-order approximation to hold is assumed (based on empirical $\alpha$ values). These speculative explanations and conditional assumptions help us develop intuitive understanding and empirical analysis of ExPO's conditions (Section 3.4 and 3.5). We believe that making bold hypotheses informed by preliminary empirical results benefits scientific understanding and discovery more than it harms. Of course, we acknowledge that these assumptions lack rigorous theoretical proofs and extensive validation (for example, how to calculate $|\Delta\theta|$ and higher-order terms, especially when our formalized $\Omega$ may not have an analytical form). However, we also believe our work presents interesting and compelling results and opens up new questions, laying a solid foundation for future, more in-depth investigations.

---

### Official Review · Reviewer_i77v · 2024-11-04

**Soundness:** 3
**Presentation:** 3
**Contribution:** 2
**Rating:** 6
**Confidence:** 4

**Summary:**

The paper introduces Model Extrapolation (EXPO), a novel method designed to expedite the alignment process for large language models. EXPO harnesses model extrapolation to substantially reduce the training overhead associated with traditional alignment methods, such as Direct Preference Optimization (DPO) and Reinforcement Learning from Human Feedback (RLHF). These traditional methods typically require substantial computational resources to fine-tune models to align with human preferences. By extrapolating weights between a supervised fine-tuning checkpoint and an intermediate, partially trained DPO model, EXPO achieves better model alignment with significantly reduced training time. The method is validated through experiments that show improved alignment performance using only 20% of the training steps required by conventional methods. EXPO consistently demonstrates performance gains across multiple open-source large language models (LLMs) on benchmarks such as AlpacaEval 2.0 and MT-Bench.

**Strengths:**

1.The paper provides theoretical explanation with comprehensive experiment results to show the eﬀectiveness of EXPO.
2. The paper presents thorough experimental results that span various model architectures and alignment techniques, highlighting EXPO’s flexibility and robustness.
3. The results suggest that EXPO could be beneficial, especially for LLMs, as it provides a computationally economical pathway to enhance alignment.

**Weaknesses:**

1. The core idea of EXPO seems to be a straightforward extension of existing model merge concept, primarily adjusting the interpolation parameter to a negative value (extrapolation). While the results are promising, this incremental shift from interpolation to extrapolation may be seen as lacking in true innovation.
2. Unlike interpolation, which typically operates within a bounded range [0,1], the extrapolation parameter α in EXPO operates within an open range [0, +∞). this open-ended range can complicate the search process, as the optimal value could vary widely depending on the model and training stage.
3. In addition to the open range above, the current manual search for α detracts from EXPO’s eﬃciency and scalability. The authors acknowledge this limitation and suggest future work on optimizing α automatically and adaptively. Having a better way to find the optimal α in this paper will make the innovation more solid.

**Questions:**

1. Given that α in EXPO operates over an open range [0,+∞), what strategies do the authors suggest for eﬀectively managing and narrowing down the search space for α?
2. The paper mentions that EXPO’s α search process takes less than 0.5 GPU hours, which appears eﬃcient. Could the authors provide a more detailed breakdown of how the search process unfolds? Specifically, how many steps or iterations are typically required, and what is the distribution of time spent in binary search versus grid search?

---

> ### Author Response · Authors · 2024-11-15
> **Rebuttal by Authors**
>
> Thank you for your review and feedback. We address your concerns and questions as follows.
>
> > Given that α in EXPO operates over an open range [0,+∞), what strategies do the authors suggest for effectively managing and narrowing down the search space for α?
>
> > The paper mentions that EXPO’s α search process takes less than 0.5 GPU hours, which appears efficient. Could the authors provide a more detailed breakdown of how the search process unfolds? Specifically, how many steps or iterations are typically required, and what is the distribution of time spent in binary search versus grid search?
>
> We answer these two questions together. We use the hyperparameter search in Table 1 as an example (the search process is actually reflected in **Figure 6 in the appendix**).
>
> * Starting with $M_2^{10\\%}$:
>   * First, with an interval of 5, we tried $\alpha=5$ and $\alpha=10$. We found both significantly outperformed $M_1$, but $(\alpha=5)>(\alpha=10)$.
>   * Then, setting the search range to $[5, 10]$ with an interval of 1, we applied binary search and tried $\alpha=7$ and $\alpha=8$. We found $(\alpha=8)>(\alpha=7)$. We then tried $\alpha=9$ and found $(\alpha=8)>(\alpha=9)$.
>   * We thus determined $\alpha=8$ as optimal (smaller search intervals might yield better results, but we deemed this unnecessary in practice).
> * Moving to $M_2^{20\\%}$:
>   * With previous experience, we first tried $\alpha=2$ and $\alpha=4$ with an interval of 2. We found $\alpha=2$ significantly outperformed $M_1$, but $\alpha=4$ performed worse than $M_1$.
>   * Then, setting search ranges to $[0, 2]$ and $[2, 4]$ with an interval of 1, we applied binary search and tried $\alpha=1$ and $\alpha=3$. We found $(\alpha=2)>(\alpha=3)>(\alpha=1)$.
>   * Next, with an interval of 0.5 in $[2, 3]$, we tried $\alpha=2.5$ and found $(\alpha=2.5)>(\alpha=2)$.
>   * We thus determined $\alpha=2.5$ as optimal. This took 5 searches total, each taking about 5min (using one A100 80GB, including inference on dev set and reward model scoring), totaling about 0.5 GPU hours.
> * For $M_2^{40\\%}$:
>   * Based on previous experience, we first tried $\alpha=0.5$ and found it outperformed $M_0$
>   * Then with an interval of 0.1, we applied grid search and tried $\alpha=0.6$ and $\alpha=0.4$. We found $\alpha=0.6$ performed worse than $M_1$ (**Note that this is a key motivation for using $[0, 0.5]$ as search range with 0.1 interval for $M_2^{100\\%}$ and models in Section 4.1**), while $(\alpha=0.5)>(\alpha=0.4)$.
>   * We thus determined $\alpha=0.5$ as optimal.
>
> Overall, we (and in practice) don't search blindly, but flexibly combine binary search, grid search, and dynamically adjusted search intervals. **These strategies are simple, practical, and represent consensus in practice.** Also note that the above search only requires **inference-level GPU hardware** (e.g., A10 24GB). Therefore, we believe that compared to the reduced training overhead (from 12 GPU hours for $M_1^{100\\%}$ to 2.5 GPU hours for $M_1^{20\\%}$) and training-level GPU hardware (from eight A100 80GB to one A10 24GB), the $\alpha$ search process in ExPO is more economical and efficient.
>
> Finally, we hope our responses can address your concerns, and we look forward to your favorable consideration of our work.

---

> > ### Comment · Reviewer_i77v · 2024-11-25
> >
> > Thank you for your careful response. I can see more details as credits for review. Maybe you could add these explanation as supplementary analysis for a more complete and rigorous work.

---

### Official Review · Reviewer_E9kQ · 2024-11-04

**Soundness:** 3
**Presentation:** 3
**Contribution:** 3
**Rating:** 6
**Confidence:** 3

**Summary:**

In this paper, the authors introduce EXPO, an efficient method for enhancing LLM alignment with human preferences. The core concept of EXPO is that the weights of a partially-trained model can serve as an intermediate result between its initial SFT checkpoint and a hypothetical, better-aligned model. By extrapolating the weights from the initial model toward the partially-trained one, EXPO can efficiently estimate the weights of a better aligned model without additional training steps. Experiments validate the improvement and data efficiency of this approach, showing consistent gains across various open-source LLMs of differing parameter sizes and multiple benchmarks.

**Strengths:**

+ The important task of efficiency of LLM alignment, and the interesting idea of extrapolating the LLM weights along the direction of the initial model toward the partially-trained one
+ Experiments are done on multimple open-source LLMs and bechmarks

**Weaknesses:**

- Lack of enough theoretical analysis
- Some parameters seem difficult to fix (e.g., \alpha)
- English should be improved

**Questions:**

- More in-depth discussion of the method is necessary (Why does it work? When does it fail? etc.), For example:  it would be helpful to discuss why extrapolation can enhance alignment while being ineffective for general training.

- Theoretical discussion is missing: there is no theoretically evidence provided to support on the factors contributing to its effectiveness

- Analysis on the selection choice and importance of \alpha across various model sizes

---

> ### Author Response · Authors · 2024-11-15
> **Rebuttal by Authors**
>
> Thank you for your review and feedback. We address your concerns and questions as follows.
>
> > More in-depth discussion of the method is necessary (Why does it work? When does it fail? etc.), For example: it would be helpful to discuss why extrapolation can enhance alignment while being ineffective for general training.
>
> We discussed this in Section 4.3. Based on our analysis in Section 3.3, alignment training typically involves smaller $\Delta\theta$, while general training (like SFT) has larger $\Delta\theta$ that invalidates ExPO's first-order approximation. Additionally, alignment training only involves adjusting the response distribution, while SFT's $\Delta\theta$ usually includes adaptation to chat templates - amplifying this part of information in $\Delta\theta$ can easily lead to model collapse and affect normal response generation.
>
> > Analysis on the selection choice and importance of \alpha across various model sizes
>
> Due to computational resource constraints, we were unfortunately unable to train larger models, but instead applied ExPO directly to open-source models (Section 4.1). We found that the value of $\alpha$ is not solely determined by model size, but is closely related to how $M_1$ was trained (based on our review of these open-source models' documentation or technical reports, such as tulu2, snorkel, starling, internlm2, etc.). The performance trends of different $M_2$ models with varying $\alpha$ are similar, as can be seen in **Figure 6 of the appendix** (the green curve for 40% training steps).
>
> Finally, we hope our responses can address your concerns, and we appreciate your favorable consideration of our work.

---

### Note · Authors · 2024-11-29

I have read and agree with the venue's withdrawal policy on behalf of myself and my co-authors.